# Consumers’ Perception and Preference for the Consumption of Wild Game Meat among Adults in Poland

**DOI:** 10.3390/foods11060830

**Published:** 2022-03-14

**Authors:** Ewa Czarniecka-Skubina, Dariusz M. Stasiak, Agnieszka Latoch, Tomasz Owczarek, Jadwiga Hamulka

**Affiliations:** 1Department of Food Gastronomy and Food Hygiene, Institute of Human Nutrition Sciences, Warsaw University of Life Sciences (WULS), 166 Nowoursynowska Str., 02-787 Warsaw, Poland; 2Department of Animal Food Technology, University of Life Sciences in Lublin, 8 Skromna Str., 20-708 Lublin, Poland; dariusz.stasiak@up.lublin.pl (D.M.S.); agnieszka_latoch@up.lublin.pl (A.L.); 3Department of Marketing and Quantitative Methods, Gdynia Maritime University, 81-87 Morska Str., 81-225 Gdynia, Poland; t.owczarek@wznj.umg.edu.pl; 4Department of Human Nutrition, Institute of Human Nutrition Sciences, Warsaw University of Life Sciences (WULS), 166 Nowoursynowska Str., 02-787 Warsaw, Poland; jadwiga_hamulka@sggw.edu.pl

**Keywords:** wild game, game meat, dishes, eating habits, adult consumer, Poles

## Abstract

Wild game meat can be a healthier, safer, and more environmentally friendly alternative to meat from farm animals. The aims of this study were to know the preferences and opinions of Polish consumers regarding game meat and its use in their diet, and to identify consumer segments based on differences in individual game meat choices, concerns, and eating habits related to game meat. The survey was conducted using the platform for online surveys among 1261 adult Poles. Six clusters characterizing the behavior of game consumers were identified (casual consumers, occasional game gourmets, indifferent consumers, occasional consumers, accidental consumers, wild game lovers) and four clusters among those who do not eat game (uninterested, restricted, dislikers, fearful). It has been found that wild game is more often eaten by hunters and their family or friends. The most common reasons for not consuming game are high prices, low availability, no family tradition, and unacceptable taste. Many positive respondents eat game because of its nutritional value but are concerned about the potential health risks and lack of cooking skills. The results of this study indicate the need for information programs for consumers about this meat. They will provide guidance to meat companies about consumer preferences for game and allow them to develop appropriate marketing strategies.

## 1. Introduction

Increasing nutritional awareness and concern for their health make consumers pay attention more and more often not only to the amount of food consumed but also to its quality, nutritional value, health benefits, and the origin of the food products. In addition, food safety, food sustainability, and reducing the climate change impact of food production are becoming increasingly important. Therefore, increasing attention is being paid to the hazards of, or the natural environment associated with, industrial livestock farming. Animal production is the cause of unfavorable climate changes and is a source of air pollution (carbon footprint). It also leads to environmental degradation and excessive water consumption. In addition, incorrect conditions on farms cause the spread of zoonoses, generating further diseases [1,2,3]. Hence, greater numbers of consumers consciously approach the issue of ethical food sourcing, including obtaining alternative raw material, as opposed to intensive livestock farming [2,4,5].

One such alternative is having a greater proportion of game meat in the overall diet. Wild game is naturally free from the undesirable traits associated with the intensive, industrial breeding of livestock. In addition, wild animals are not exposed to the stress associated with industrial breeding, and, when correctly hunted, do not have the stress associated with the road to the slaughterhouse. Their meat contains only trace amounts of adrenaline. Varied foods consumed in the natural habitats of wild animals, as well as higher physical activity, are just some of the factors influencing the generally desirable specific taste, appearance, texture, and unique nutritional properties of game meat.

Meat obtained from the carcasses of wild animals is usually considered a culinary delicacy. However, its quality varies greatly due to inter- and intraspecific differences [6,7]. In Europe, including Poland, there are various species of game animals, which are traditionally divided into large wild game (e.g., elk, deer, fallow deer, wild boar, roe deer) and small wild game (e.g., hare, game birds: pheasant, partridge). Due to the climatic differences, geographical conditions, and the composition of food (grasses, herbs, agricultural crops), the range of variability in the chemical composition, physical properties, and sensory quality of game meat is quite wide. Thus, the nutritional value of game meat depends on the species, sex, age, condition, and physiological condition of the animals, as well as the hunting season (e.g., accumulating fats for the winter), the foraging area, climate, and the particular part of the carcass [7,8,9,10,11].

Overall, it is meat with a relatively low energy value; a high level of wholesome, easily digestible total protein; and low fat and cholesterol content [7,8,9,10,11,12,13,14,15,16]; with a good fatty acid composition and *n*-6 to *n*-3 ratio (lower or close to 4) [4,7,8,15,17,18,19,20,21]. In addition, it has good healthy lipid indices such as the atherogenic index (AI) and thrombogenic index (TI) [15,21]. It has been shown that the amounts of essential amino acids, vitamins (PP, B_1_, B_2_, B_6_, E) [7,13], micro- and macro elements (phosphorus, magnesium, iron, manganese, and zinc) [13,17] in game meat are much higher than in the meat from farm animals. Game can also be viewed as a source of bioactive compounds: among others, conjugated linoleic acid (CLA), as well as carnosine and anserine [22,23,24,25,26].

Despite the numerous advantages of wild game (nutritional, taste, health), its consumption in many countries in Europe, including Poland, Great Britain, Czech Republic, Croatia, Germany, Norway, and Sweden, is low. In recent years, it was from 0.2 to 1.1 kg/person/year [27,28,29,30,31,32], but only 2–4% of the population consumed this type of meat regularly [33]. Greater consumption was found among hunters and their families [34,35].

The reasons for the low consumption of game can be found, among others, in some consumer concerns about zoonoses. The meat of game animals, especially wild boars, is considered as the main source of *Trichinella spiralis* in Europe [35,36,37,38], and the meat of deer, hares, and game birds poses a risk of infection with toxoplasmosis. There may also be a risk of *Yersinia enterocolitica* [39,40,41], *Toxoplasma gondii* [42,43,44,45], *Salmonella* [45,46,47], dioxins and PCBs [48], and the potential hazards of heavy metals due to the progressive degradation of the natural environment and their accumulation in the feed of wild game [37,49,50,51]. Due to the presence of environmental pollutants in the food chain of wild animals, and the presence of parasites and infectious diseases, meat obtained from these animals should be thoroughly tested. However, many authors [32,37,52] argue that these fears are unfounded. In many respects, game is a decidedly safer meat than livestock meat, especially in countries where restrictive veterinary regulations are in force, and levels of harmful compounds do not exceed acceptable limits and do not constitute a risk to consumers. Barriers to wild game consumption also include: high price, difficulty of access to the product [27,28,29,30,31,32,33,34,35,36,37,38,39,40,41,42,43,44,45,46,47,48,49,50,51,52,53], limited supply [27,54], health safety concerns [55,56,57,58,59], specific taste and smell, lack of the habit of consuming it, as well as an inability to prepare dishes properly [27,54,60,61,62,63].

Although the high nutritional and health-promoting value of wild game has been documented, with its positive impact on the health and functioning of the human body [14,64], there are a limited number of studies on consumption, preferences, and the determinants of attitudes towards wild animal meat among consumers. Current data on the preferences of game consumers are of great importance for shaping the environmental (concerning the natural environment) and food policy (regarding the protection of raw materials for the food industry) not only in Poland, but also in countries with a similar climate, fauna, and flora as well as culinary traditions.

Hence, to fill the aforementioned scientific gaps, the aims of this study were:(i).to know and understand the preferences and opinions of Polish consumers regarding wild game meat and its use in their diet;(ii).to establish attitudes toward wild game meat/dishes’ consumption;(iii).to identify, describe, and compare consumer segments based on differences in individual food choices, concerns, and eating habits related to wild game meat.

## 2. Materials and Methods

### 2.1. Study Design and Participants

This cross-sectional survey was designed as a study with a convenience sampling and conducted among adults living in Poland, using the Google Forms web survey platform. The link to the online survey was shared through social media, such as Facebook, Instagram, and WhatsApp, and by personal contacts of the research group members. We also asked the participants to share the study link to increase the number of persons who received the invitation to the study and thus increase the study participants. This kind of investigation allowed us to conduct a nationwide survey, especially during the pandemic, which limited the opportunity to conduct stationary studies involving respondents. Participants received information about the anonymity of the study, the voluntary nature, and the possibility to stop their participation at any study stage. Moreover, a questionnaire provided on a webpage increased the sense of anonymity and gave an opportunity to participate in the study at a time convenient for the respondent.

The study was carried out in accordance with the Declaration of Helsinki [65] and participation was entirely voluntary. Respondents did not provide their names or contact information (including the IP address), and could finish the survey at any stage, according to the General Data Protection Regulation of the European Parliament [66]. Considering the anonymous nature of the online survey and the inability to track the sensitive personal data of respondents, the survey did not require the consent of the ethics committee (as well, written informed consent for the study was not required).

The survey was conducted on a group of 1261 adult respondents in Poland. Inclusion criteria for respondents of the study were as follows:−aged over 18 years,−interested in participating in the study,−living in Poland,−Internet access.

Each respondent who agreed to participate in the survey was invited to complete the questionnaire with data according to the best of their knowledge.

Initially, a linear snowball sampling approach with a convenience sample of the initial subject was applied [67]. Because of the relatively low response among the target population, the questionnaire was also delivered to closed groups such as foresters and hunters. The exclusion criteria for respondents of the study were as follows:−respondents under the age of 18,−duplicate responses,−questionnaires with missing or inconsistent data.

The final sample size for analysis was 1251 as 10 responses were excluded. After verification, the final data set included 726 wild game consumers and 525 non-consumers.

### 2.2. Questionnaire

The questionnaire was designed based on previous studies of consumer eating habits regarding wild game meat consumption [56,57,59,60,63,68,69,70,71]. The questionnaire was checked by means of a pilot study with 20 people, and any problems were identified, and the questionnaire amended. The pilot test indicated that completing the form would take each participant around 10 min.

The questionnaire consisted of two parts, with the first part containing 11 questions relating to consuming wild game (Appendix A). The questions covered habits associated with the use of wild game such as choosing the type of wild game, circumstances of consumption, type of meals and culinary methods used for preparing them at home, sources of wild game meat, frequency of and reasons for wild game consumption or non-consumption, as well as concerns about consuming this type of meat. The second part of the questionnaire included five questions related to the respondents’ sociodemographic details (gender, age, education, dwelling place, financial situation).

Before starting the analysis of the survey results, the internal consistency and reliability of the survey questionnaire structure were checked. For this purpose, the Cronbach’s alpha test was used. The value of the alpha coefficient for the entire questionnaire (all questions in the survey) was 0.79. This result indicates the satisfactory consistency of the survey questionnaire and allows the obtained results to be used in a further analysis.

### 2.3. Data Analysis

A statistical analysis of the results was performed using Statistica software (version 13.3 PL; StatSoft Inc., Krakow, Poland).

The chi-square test was used in the study to assess the influence of factors describing the population on the examined features. The significance of differences between the values was determined at a significance level of *p* ≤ 0.05.

The Mann–Whitney U test was used to assess the influence of eating/not eating wild game on sociodemographic factors. The null hypothesis was the lack of significant differences between the distributions of factors in both groups; the alternative hypothesis-significant differentiation of factors caused by eating wild game. The significance of the test was assumed at the level of *p* ≤ 0.05.

The aim of the analysis was to create groups of respondents with a similar approach to game consumption. The respondents were divided into two groups: people who eat, and those who do not eat, wild game. Homogeneous groups were created among people consuming this type of meat in terms of the purchase and consumption of game. In the non-game group, the respondents were divided according to the reasons why they did not eat game meat.

The vast majority of the variables in the study are on an ordinal scale, with a few variables on the nominal scale, and two on the ratio scale. For this reason, a cluster analysis was used to divide the respondents. Due to the qualitative nature of the variables, the analysis used the percentage discrepancy as a measure of distance, being the quotient of the number of dimensions with inconsistent values and the number of all dimensions. To study the distances of clusters, complete linkage clustering was used, i.e., the distance of the farthest elements of both clusters.

In the study of game eaters, due to the large number of variables, it was decided to limit these by grouping them using the agglomeration method, and leaving one variable from each group in the study. This procedure reduced the number of variables used to eight without any significant loss of information. The system of variables reduced in this way was used to create the division of cases (respondents). The optimal number of clusters was determined by the agglomeration method, and then by the *k*-means method, and all consumers were divided into homogeneous clusters. Due to the difficulties in interpreting the obtained results, it was decided to create the smallest possible number of clusters, ensuring a clear segmentation of the community. For each of the clusters, the medians were calculated and used to identify the characteristics of the groups. The analysis for non-eaters was immediately limited to case segmentation and descriptions of the obtained clusters.

## 3. Results

### 3.1. Characteristics of Respondents

The characteristics of the respondents are presented in Table 1.

The study involved mainly women, with secondary or higher education, living in a variety of dwellings. The respondents were in the range between 18 and over 51 years old, had access to a computer and the Internet, and had computer literacy skills. In the study group, 726 respondents consumed and 525 did not consume game. A fairly large group of game consumers (26.2%) was represented by hunters, most of whom (approx. 79%) had hunting experience of over 6 years.

### 3.2. Reason for Consumption and Non-Consumption of Game Meat by Respondents

The consumption of game meat and its preserves (*n* = 726) was most often justified by the taste of game (68.6% of responses) and its health properties (38%), as well as family traditions (29.9%), availability of such meat (29.8%), participation in hunting, and the need to use the obtained meat (20.5%). Other reasons included game fashion (6.1%), popularity of this game (4.7%), and allergy among family members (2.8%); as well as other reasons such as the originality of the taste, the willingness to try new tastes, and the need to use game received as a gift (4.7%).

The most common reasons for not consuming game (*n* = 525) were the lack of any tradition of eating this meat (44% of the answers), low availability (41.7%), high price (37.7%), and fear of diseases (32%). About 24% did not eat game for ethical reasons, or were vegetarians and wanted to care for the planet. Two people indicated religious considerations and disgust (2.5%). They also indicated unacceptable taste of the meat (23.8%), lack of skills in preparing a tasty dish on your own (21.1%), and other respondents “believe that good wild game dishes can only be eaten in the restaurant” (4.6%). The lack of being convinced that this meat is healthier than other meat (19.8%), as well as health problems (allergies) limiting game consumption, were also mentioned.

Most of the respondents (72.7%) who ate game had no fear of eating it, as they most often knew the origin of the game. On the other hand, the remaining consumers were afraid (10.2%) or sometimes had a fear (17.1%) of eating this meat. The respondents were concerned about the low quality of meat and zoonoses, mainly trichinosis, and the risk of food poisoning, especially in the case of meat from unknown sources. They were concerned that the meat was not properly tested and prepared. Two people indicated ethical doubts related to the methods of obtaining game meat.

The reasons for consumption or non-consumption of game meat depended on the sociodemographic data (*p* ≤ 0.05). Due to tradition, game was eaten more often by people living in a village (*p* ≤ 0.001), and due to culinary skills, by people over 51 years old (*p* ≤ 0.001). People aged 31–40 years (*p* = 0.042) with a higher education (*p* = 0.021) had greater concerns about game meat. In turn, women, people aged 31–40 years, with a higher education, and living in cities with more than 500,000 residents had significantly more often given up eating game for ethical reasons (*p* ≤ 0.05).

It can be assumed that eating or not eating wild game can significantly differentiate the respondents. To verify this, the chi-square test was performed for the influence of the gender of the respondents (feature on a nominal scale) on game consumption, and the Mann–Whitney U test for the remaining sociodemographic factors (feature on an ordinal scale). The results of the Mann–Whitney U test and median values in both groups are shown in Table 2.

Based on the chi-square test (χ^2^ = 42.341, *p* = 0.000), it can be concluded that gender significantly differentiates the propensity to eat wild game. Similarly, on the basis of the Mann–Whitney U test, this can be stated for age, education, and financial situation. People who do not eat game are mainly younger women, aged 18–30, with a secondary education. The average game eater is a middle-aged man (31–40 years old) with a university degree. The financial situation in both groups is similar (the medians have the same value), but people who eat game assess their financial situation on average slightly better.

### 3.3. Habits Related to Wild Game Consumption

The respondents ate game meat at different frequencies: half of them—less than once a year (*n* = 366, 50.41%), next—once every 2–3 months (*n* = 118, 16.25%), and then two-three times a month (*n* = 117, 16.12%). The remaining respondents consumed it three or four times a week (*n* = 68, 9.37%) or less frequently, i.e., once a month (*n* = 31, 4.27%) and once a week (*n* = 26, 3.58%). Significantly more often, it was consumed once a week or several times a week by men (*p* ≤ 0.001), people aged 18–30 years, with higher education (*p* = 0.038), living in a village (*p* = 0.002), or with a good or very good financial situation (*p* ≤ 0.001).

The respondents mentioned that the most common places to consume game meat were at home (*n* = 356, 49.0%), with friends (*n* = 317, 43.7%), or in catering establishments (*n* = 271, 37.3%). The choice of the place of consumption depended on gender and age (*p* < 0.05).

Education and dwelling place did not influence the choice of where to eat game (*p* > 0.05). About 30% (*n* = 208) of the respondents prepared it themselves at home. Men significantly more often declared this (*p* ≤ 0.001), people aged 31–40 and over 51 years (*p* ≤ 0.001), and mainly hunters with extensive experience. Young people aged 18–30 years significantly more often declared game meat consumption at home (*p* = 0.007) or in catering establishments (*p* = 0.017). Other places to eat game that were mentioned (*n* = 72, 9.9%) included tastings at fairs, culinary competitions, or recording tv programs with tastings.

The respondents most often obtained game from a hunter (friends and family (*n* = 351, 48.5%). Some of them hunted (*n* = 181, 24.9%); while others used shops/specialized wholesalers (*n* = 73, 10.1%); or bought in a typical shop, supermarket, or hypermarket (*n* = 43, 5.9%). Several people (*n* = 10, 1.4%) mentioned other sources of game acquisition, such as markets, fairs, and gifts from hunters.

Women significantly more often bought it in typical stores such as supermarkets and hypermarkets (*p* = 0.039) or obtained it from hunters (*p* = 0.003), while men, people aged 31–40, with higher education, with a good or very good financial situation (*p* ≤ 0.001), and those living in a village (*p* = 0.007) hunted themselves.

The respondents declared that they most often ate wild boar (86.2%), wild game birds (72.3%), and roe deer (70.9%). In addition, a smaller percentage of respondents mentioned hare (58.7%), deer (55.5%), and others (2.58%) such as fallow deer, moose, and wild rabbit.

Game dishes were prepared and eaten most often as the main course (39.8% on average) and as cold cuts and other preserves (27.4% on average). According to the Polish tradition, the hare is most often used to prepare pates, which was also confirmed in this research. Other methods of preparation, such as appetizers, soups, and ingredients of other dishes, were used much less frequently. Based on this, it can be argued that game meat is considered a valuable raw material in the main course, and processing into cold cuts and other preserves allows for the safe storage of meat and protects against losses. The table below presents detailed data on the consumption of game dishes (Table 3).

Among the culinary methods used or chosen by the respondents, the most frequently mentioned were stewing (70.4% of answers), baking (70.1%), cooking (64.9%), smoking of products (67.4%), preparing non-smoked products (60.9%), frying (60.7%), grilling (53.2%), consumption as raw, e.g., ”wild tartare” (44.5%), and others (8.3%).

People who declared that they prepared game dishes on their own indicated that they have used stewing, baking, and preparation of non-smoked products from time to time. Frying, grilling, and preparation of non-smoked preserves were used less frequently. The frequency of consumption of raw game meat was low.

The choice of the method of game meat preparation depended on the gender, age, and dwelling place (*p* < 0.05) of the respondents. However, the education and financial situation of the respondents did not affect the way of preparing game meat (*p* > 0.05). Women significantly more often declared that they used the process of cooking (*p* = 0.002), stewing (*p* ≤ 0.001), grilling (*p* ≤ 0.001), and preparing non-smoked products (*p* = 0.00001). However, men chose frying (*p* ≤ 0.001), baking (*p* = 0.003), eating as a raw (“tartare”) (*p* ≤ 0.001), and preparing smoked products (*p* ≤ 0.001). People over 51 years of age significantly more often declared preparing stewed game (*p* ≤ 0.001), and people living in a village chose the frying process (*p* = 0.021).

### 3.4. Clusters of Wild Game Consumers and Non-Eaters

In the analysis of game eaters, 69 variables describing the attitude of game eaters to this meat and its products were used (answers to questions 3–10 in the questionnaire). In order to reduce the number of variables, the agglomeration method of cluster analysis was used. Its results are presented in Figure 1a. The cut-off level of linkage was set at 0.7. As a result of agglomeration, all variables were divided into seven describing groups: frequency of consumption of wild boar meat, frequency of consumption of other types of game, frequency of consumption of traditionally prepared game dishes, frequency of consumption of less traditionally prepared meat, frequency of self-preparation of game, reasons for consuming game, and concerns about its consumption.

The dimensions described in this way were used to divide the respondents into groups. There were 726 respondents declaring game consumption. Initially, the agglomeration method was used to determine the optimal number of groups. Taking the linkage distance equal to 0.9 as the border value, the division into six clusters was established. The agglomeration results are presented in the diagram (Figure 1b).

Using the k-means method, the respondents were finally divided into six clusters. Medians calculated for previously constructed dimensions and sociodemographic variables were used to describe the respondents in the clusters (Table 4).

As a result of the analysis of average values for the previously indicated dimensions and variables describing the respondents, descriptions of people’s characteristics for the created clusters were obtained:**Cluster 1** (Casual consumers, *n* = 107, 15%)—A person who occasionally consumes game preserves, more often from wild than from other animals, does not come into contact with unprocessed meat from wild animals too often, and does not cook by himself. Like everyone in the other clusters, he is not afraid of eating wild boar meat. A person who does not shy away from game but does not seek it. The representative of this group is a middle-aged man (30–40 years old) with a higher education from a small town.**Cluster 2** (Occasional game gourmets, *n* = 240, 33%)—Occasional eaters of game, which is often in the form of a main course of wild boar and cold cuts made of less popular types of meat, and they are gourmets. He often prepares game dishes himself. The average respondent is a well-educated, middle-aged man (30–40 years old) from the city.**Cluster 3** (Indifferent consumers, *n* = 70, 10%)—A person who has contact with game, but sporadically. He remembers traditional wild boar dishes. In addition, game is not eaten or prepared. The representative of this group is a young man (18–30 years old) with a secondary education from a big city.**Cluster 4** (Occasional consumers, *n* = 95, 13%)—A person who has tried mostly preserves of game meats. It can be assumed that this contact was accidental and definitely sporadic. A person indifferent to game. The average respondent is a young woman (18–30 years old) with a secondary education from a big city.**Cluster 5** (Accidental consumers, *n* = 138, 19%)—A person who declares that she has tried meat and game preserves, but it was a one-time contact. She is not interested in game. She does not want to make it or eat it. Perhaps this attitude is based on ethical considerations; although, like others, she appreciates the taste of venison. A person with a negative attitude towards game. The cluster is represented by a young woman (18–30 years old) with a higher education from a small town (less than fifty thousand residents).**Cluster 6** (Wild game lovers, *n* = 76, 10%)—A person who has frequent contact with game and is interested in game. She eats both wild boar and less common types of meat. She prefers traditionally prepared wild boar dishes, but occasionally uses dishes prepared in a different way. She often prepares game dishes herself. The representative of the cluster is a middle-aged woman with a higher education from a small town.

Ten variables (answers to question 2 in the survey questionnaire) were used in the analysis of the causes influencing the non-consumption of game meat. All variables were dichotomous. The sample contained 525 observations. The agglomeration method of a cluster analysis was used to determine the optimal number of respondent groups. As a result, assuming the boundary level of the linkage at the level of 0.8, it was found that the optimal number of groups was four. The results of the agglomeration are shown in Figure 2.

Grouping using the *k*-means method made it possible to divide the respondents into the above-mentioned groups. The values of dominants and medians in clusters for the variables forming the groups and the variables describing the respondents are presented in Table 5.

On its basis, people who do not eat wild meat can be described in the form of four clusters. The average representative of each cluster can be described as follows:**Cluster 1** (Uninterested, *n* = 197, 38%)—People who answered all questions negatively. These respondents did not consider why they did not eat game. Most likely they have no contact with it and will not think to try it. The average representative of this group of people is a young woman (18–30 years old) with a secondary education, living in a small town, with a good financial situation.**Cluster 2** (Restricted, *n* = 82, 16%)—People who do not eat game for technical reasons. They consider it too expensive and inaccessible, and the lack of tradition of eating this type of meat does not encourage them to overcome these difficulties. The representative of this group is a mature woman (41–50 years old), having a secondary education, living in a small town. She assesses her financial situation as not the best.**Cluster 3** (Dislikers, *n* = 162, 32%)—People in this group do not eat game due to the lack of certainty about the greater taste of game. They are indifferent to other factors, but they do not find a reason to eat it. The average respondent in Cluster 3 is a middle-aged woman (31–40 years old) with a secondary education, having a good financial situation, and living in a large city.**Cluster 4** (Fearful, *n* = 73, 14%)—The last group are people who do not eat game due to concerns about the safety of this type of meat. For example, they are concerned about the possibility of contracting trichinosis. They are indifferent to other factors. The representative of this group is a young woman (18–30 years old) with a secondary education, living in a small town, and having a good financial situation.

Among the respondents, 127 (24% of the total) said that they do not eat game for ethical reasons. A strong influence of this factor on the attitudes of consumers was expected. However, these people did not become an essential part of any of the distinguished clusters. Ethics is not a factor that strongly separates the analyzed community. This factor usually occurred also with other reasons for not accepting this meat, and they distinguished their clusters more strongly. Perhaps the ethical values were merely declarative values, and the reasons for not eating game were, in fact, different.

## 4. Discussion

### 4.1. Reasons for Consuming and Not Consuming Game Meat by Respondents

Respondents who consumed game mostly liked its taste, appreciated its health values, and had easy access to it due to the fact that they were hunting themselves or someone from their family or friends was hunting, and there was a need to use the obtained meat. They also had family traditions and habits of eating this meat. Other authors, similar to this study, have also indicated that hunters and their family members or their friends eat game meat more often [33,34,35] because of a need to use the obtained meat.

As in this study, other authors also emphasized that the most common reasons for not consuming game meat are low availability and lack of tradition in the family [27,59,71], as well as an unacceptable taste of this meat and its preserves [4,27,54,60,61,62,63,71]. As can be seen from previous studies, one’s habits and own experiences [72,73,74], knowledge of the product [63], and general knowledge about game meat and the possibility of using it [75], together with the desire to diversify the diet [59], have the most significant influence on consumer decisions regarding the consumption of game.

Similar to this study, many authors cited taste and other sensory characteristics such as tenderness and juiciness as important reasons for consuming game [4,59,61,63,68,76,77,78,79]. The taste of game is specific and is both an advantage and a disadvantage of this meat. Those who accept it willingly and often eat game declared their willingness to eat it in the future [63]. These were usually people with a higher education, aged 20–40 years, and with a high income [71]. The taste and aroma were also perceived as factors that make game less attractive [34]. Due to the lack of knowledge and skills in its preparation, consumers did not prepare dishes from this meat, thinking that it would be unpalatable. The unknown taste of game was more often an incentive to try it for men up to 30 years of age and people with a primary education [71]. It should be noted that the sensory value, nutritional value, and natural origin were perceived as the main value of game [4,5,56,60,63,78,80,81,82].

The respondents who did not consume game indicated the high price of this meat. Game is commonly regarded as an expensive, delicatessen/exclusive meat, as well as an “ecological” product, which is related to the specificity of obtaining meat, the nutrition, and the seasonality of its occurrence [4,27,54,59,60,63,71]; livestock meat is much cheaper [59]. British consumers often chose to buy game meat locally due to its lower price because of the shorter distribution network [54]. Latvian consumers pointed out that the motivation to eat game would be helped by making game prices equal to pork and beef prices, and to increase the availability of this meat [80]. A greater tendency to pay for hunted wild game meat characterized consumers with a positive attitude to hunting, with the highest level of knowledge about wild game, the meat, and its products [13,56,61,63,74,83].

The increase in the share of game in the consumer’s diet due to its health properties, environmentally sustainable production [5], as well as high quality of meat [77], as in this study, is also indicated by other authors [4,59,60,63,71,84,85]. The safety of game meat consumption is an important issue that determines respondent interest, which is confirmed by many authors [57,71,82,86,87].

It is estimated that many human infectious diseases come from wild animals, so the concerns regarding game are justified, as evidenced by various disease outbreaks around the world [88,89]. On the other hand, it is emphasized that these concerns are often unfounded. The statistics from the European Food Safety Authority show that the number of confirmed cases of, for example, trichinella is small. In general, game is considered a safe product when all sanitary and safety rules are followed [20]. Consumers, as in this study, are most often concerned about zoonoses [4,54,55,56,58,71,88,90], and microbiological contamination [91]. The authors of the above-mentioned studies indicated that consumers are aware that meat purchased from a proven source, veterinary tested, and subjected to appropriate heat treatment is safe for consumption. In European countries, the microbiological quality of game has significantly improved as a result of good hunting practices and good hygiene practices [92], as well as appropriate sanitary and veterinary supervision over game meat. The level of consumer protection against trichinosis in the European Union is high [93]. Trichinosis is diagnosed mainly among hunters, their families, and friends. The main reasons for the identified cases are eating meat that was not subjected to veterinary control or appropriate culinary treatment [94,95]. The way of perceiving the safety of game depends on the level of nutritional knowledge, consumers’ experiences [60], and knowledge of the origin of the meat, which for consumers is a guarantee of safety [13].

According to many Polish consumers, game meat is perceived as complicated and time-consuming to prepare, and requires knowledge of its specificity, and appropriate skills in preparing tasty game dishes [60,63]. Latvians [80] emphasized that if they had knowledge and recipes of how to prepare game, it would increase their motivation to buy this meat. Negative consumer experiences of cooking game meat may result in non-acceptance [60], while positive experiences may be seen as an opportunity to demonstrate culinary skills [56].

Among the group of respondents who did not consume game meat, there were those who pointed to the ethical aspects. They were mainly vegetarians who did not approve of killing animals. In the minds of many consumers, hunting is solely equated with killing animals, and not a way to maintain the biological balance in the environment [61,96]. Farmed game can be an alternative; however, research shows that wild and farmed game meat have different nutrient contents [7]. Some consumers find wild game meat more animal welfare friendly as it is considered more ethically justified than farmed meat because the animals are free until they are killed [61,97]. Another reason game meat is not consumed is that consumers have never tasted it before [59].

### 4.2. Habits Related to Wild Game Consumption

Half of the respondents ate game less than once a year. Those who ate more often hunted themselves or had a hunter in their family and lived in villages or small towns. Other authors [63,71] also indicated the low frequency of game meat consumption by consumers, especially women [98]. Greater interest in the consumption of game meat was shown by the inhabitants of rural areas [59,74,99], although in some studies it was indicated by inhabitants of cities [71]. The frequency of eating game meat increased with the level of education. Consumption several times a month was declared more often by people with higher and secondary educations [71]. Despite the fact that respondents declared low game consumption, it was commonly believed that they ate too little of it, and they, therefore, declared their readiness to eat it, especially men, inhabitants of large cities, the educated, and those aged 31–40 years [71]. The respondents had most often obtained game from hunters and from the catering establishment, and less often from a shop or a specialized wholesaler. Other authors also paid attention to the choice of where to buy game meat [80,100,101]; according to the consumers, they bought their game mainly in specialized stores, and rarely in supermarkets, agricultural markets, or restaurants. Sourcing farmed game meat directly from the producer was less popular [80,101].

Game was rarely eaten raw. Preferences for the consumption of game after heat treatment and in the form of preserves, in particular wild boar, deer, and roe deer meat, and to a lesser extent, hare and elk, were also indicated by other authors [59,80].

Taking into account the results of Tomasevic et al. [56] in studies conducted in ten European countries, game consumption was mainly influenced by geographic location. Populations in the countries of Southeast Europe consumed more of it than in the countries of Central Europe. It was also influenced by the age and gender of the respondents, and male and older consumers ate more game than women and younger consumers, which was also confirmed in our study.

In the identified “Occasional game gourmets” and “Casual consumers” clusters, comprising 48% of respondents consuming game meat, men aged 30–40 years were dominant, and, therefore, younger than in the studies by Tomasevic et al. [56]. Young women (18–30 years old) and women under 40 years were less interested in eating game. This structure of consumers can be explained by the traditions of food consumption by men, the high level of acceptance of its palatability, and the fact that men tend to hunt.

According to Tomasevic et al. [56], consumers recognized the health and nutritional benefits of game, considering it to be more organic than farmed meat. Central European consumers, especially the younger generation, were more interested in the price, quality, and sensory features (taste, smell) of game. The research by these authors [56] shows that almost half of them (44.1%) bought game directly from hunters, and 12.2% were hunters themselves. Consumers in Southeast Europe ate game mainly on social occasions, while consumers in Central Europe ate game mostly for hedonistic purposes. Bulgaria has a particularly rich culinary tradition of eating game.

Our research also shows that the traditions of eating game in Poland are not properly cultivated. The consumer clusters identified in this study are similar to the results of Niewiadomska et al. [57], in which three groups of consumers were identified as selective, indifferent, and fearful. Differences between groups of game consumers may result from the consumers’ own experiences, habits from the family home, and factors such as the influence of the environment [57], as well as the desire to reduce the consumption of animal products in favor of plant products and sustainable development [3].

### 4.3. Limitation

The strength of our study is the relatively large representative sample of adult Poles.

We are aware that in the case of people who do not eat game meat, our group was not a representative population for the entire adult population in our country (*n* = 525), which may be a barrier to this study. However, it came from all over Poland, and its selection took into account all determinants also included in the group of game meat consumers. Therefore, it can be considered as a reference group for comparative studies. Using an online survey to collect data can be considered a benefit as it allows the possibility of reaching a larger group of people from different backgrounds, which was very valuable when collecting data during a pandemic. On the other hand, it was also a limitation resulting from the possibility of participation only by people with Internet access (eliminating mainly the elderly). Moreover, the convenient selection of respondents used in the study, and not allowing for the consumption of farmed game, may slightly falsify the results and cannot be generalized to the entire population.

Finally, some bias may result from the characteristics of the survey method that we applied (a cross-sectional approach). It should also be emphasized that our findings are specific to the Polish cultural background and should be treated with caution in relation to other countries. Despite the limitations, the results obtained are of practical importance and accurately demonstrate the habits of consumers related to consuming wild game meat in Poland.

## 5. Conclusions

In recent years, consumers’ interest in wild game has increased, largely because they appreciate its nutritional and health-promoting values; however, many people still have numerous concerns, including ethical ones. Moreover, considering the high price of wild animal meat as well as its low availability (mainly in specialist stores and in hypermarkets—only seasonally, mostly around Christmas), there are few studies describing a comprehensive approach to the preferences of wild game consumption in Poland. Increasing game consumption requires changing consumer habits, creating a healthy lifestyle trend, testing the quality and nutritional value of game, proper advertising of this meat, and its certification.

This study will allow meat industry representatives to better understand the game market by providing information on consumers’ attitudes and habits towards game. It will also help to develop effective game meat marketing strategies to increase its consumption. Our results can be used as a basis for debate to broaden and better understand game meat perception and use by a wider group of consumers.

## Figures and Tables

**Figure 1 foods-11-00830-f001:**
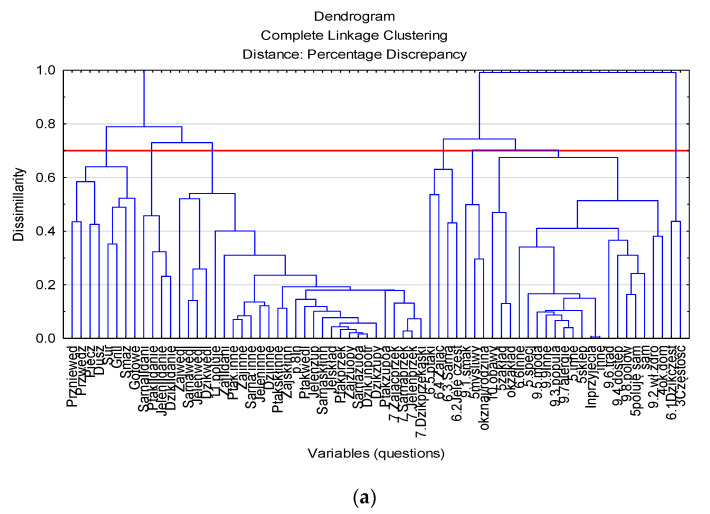
Results of agglomeration of cluster analysis for variables (**a**) and cases (**b**).

**Figure 2 foods-11-00830-f002:**
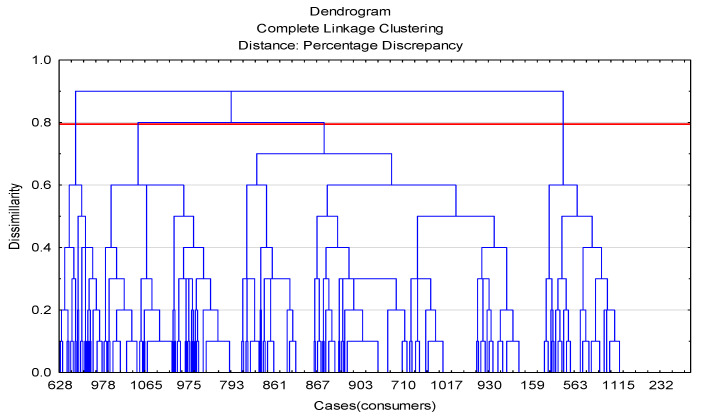
Case agglomeration results for non-game eaters.

**Table 1 foods-11-00830-t001:** Characteristics of the surveyed sample of respondents.

Population Features	Group	Respondents			
		Eating Wild Game		Not Eating Wild Game	
		Number *n*	Percentage %	Number *n*	Percentage %
Total	All respondents	*n* = 1251		100%	
	Respondents divided into group of eating or no eating wild game	726	100 (58 *)	525	100 (42 *)
Gender	Women	379	52.2	370	70.5
	Men	347	47.8	155	29.5
Age	18–30 years old	287	39.5	268	51.1
	31–40 years old	147	20.2	84	16
	41–50 years old	129	17.8	106	20.2
	>51 years old	163	22.5	67	12.7
Education	Vocational or primary school	22	3	34	6.5
	Secondary school	264	36.4	315	60
	Higher education (university)	440	60.6	176	33.5
Dwelling place	Village	245	33.8	164	31.2
	City up to 50,000 inhabitants	136	18.7	106	20.2
	City of 50,001–100,000 inhabitants	64	8.8	79	15
	City of 100,001–500,000 inhabitants	141	19.4	99	18.9
	City over 500,000 inhabitants	140	19.3	77	14.7
Financial situation in own opinion	Very good	131	18	54	10.3
	Good	408	56.2	178	33.9
	Not good, not bad	178	24.5	255	48.6
	Bad and very bad	9	1.3	38	7.2
Game consumers	Hunters	190	26.2	−	−
	Other consumers	536	73.8		
Hunters	Hunting 0–5 years	40	21.1 **	−	−
	Hunting 6–10 years	43	22.6 **		
	Hunting over 11 years	107	56.3 **		

* percentage of the total group, ** percentage of the group of hunters, − not applicable.

**Table 2 foods-11-00830-t002:** Values of test statistics and significance *p* in the Mann–Whitney U test and median values in the group of people consuming and not consuming game.

Population Features	Mann–Whitney U Test		Medians	
	Test Statistic U	*p*	No Eating Wild Game	Eating Wild Game
Age [years]	164,762.5	0.000	18–30	31–40
Education	137,940.0	0.000	secondary school	higher education
Dwelling place	187,441.5	0.609	city up to 50,000 inhabitants	city up to 50,000 inhabitants
Financial situation in own opinion	142,491.0	0.000	good	Good

**Table 3 foods-11-00830-t003:** Wild game consumption and meals prepared from it.

Wild Game	Sources	Respondents (*n* = 726)	
		Number (*n*)	Percentage (%)
Wild boar	Appetizers	73	10.1
	Soups	26	3.6
	Main course	350	48.2
	Cold cuts and other preserves	294	40.5
	Ingredients of other dishes	5	0.7
	Other	61	8.4
Deer	Appetizers	37	5.1
	Soups	35	4.8
	Main course	230	31.7
	Cold cuts and other preserves	160	22
	Ingredients of other dishes	16	2.2
	Other	103	14.2
Roe deer	Appetizers	45	6.2
	Soups	8	1.1
	Main course	388	53.4
	Cold cuts and other preserves	192	26.5
	Ingredients of other dishes	37	5.1
	Other	55	7.6
Hare	Appetizers	59	8.1
	Soups	11	1.5
	Main course	146	20.1
	Cold cuts and other preserves	298	41.1
	Ingredients of other dishes	69	9.5
	Other	51	7
Wild game birds	Appetizers	15	2.1
	Soups	74	10.2
	Main course	332	45.7
	Cold cuts and other preserves	52	7.2
	Ingredients of other dishes	78	10.7
	Other	32	4.4

**Table 4 foods-11-00830-t004:** Medians of dimensions describing the behavior of respondents and sociodemographic variables in the proposed clusters.

Behavior of Respondents/Sociodemographic Variables	Cluster 1	Cluster 2	Cluster 3	Cluster 4	Cluster 5	Cluster 6
Number of respondents (*n*)	107	240	70	95	138	76
Wild game consumption ^A^	3	3	2	2	2	4
Frequency of eating different game ^A^	2	2	2	2	2	3
Traditionally cooked game ^B^	0	3	3	0	0	3
Less commonly prepared game ^B^	4	4	0	4	0	0
The frequency of preparing game on its own ^C^	3	4	1	0	0	4
Reason for eating game-taste ^D^	2	2	2	2	2	2
Concerns about game meat ^E^	1	1	1	1	1	1
Gender ^F^	1	1	1	2	2	2
Age ^G^	2	3	1	1	1	2
Education ^H^	3	3	2	2	3	3
Dwelling place ^I^	2	2.5	3	3	2	2
Financial situation ^J^	3	3	3	3	3	3

^A^ I did not use (1), more rarely than once a year (2), once every 2–3 months (3), once a month (4), two-three times a month (5), once a week (6), three or four times a week (7); ^B^ not at all (0), appetizers (1), soups (2), main course (3), cold cuts and other preserves (4), ingredients of other dishes (5), other (6)**;** ^C^ none (0), least often (1), rarely (2), moderately frequently (3), quite often (4), the most often (5); ^D^ no (1), yes (2); ^E^ yes (3), sometimes (2), no (1); ^F^ women (2), men (1); ^G^ 18–13 years (1), 31–40 years (2), 41–50 years (3), over 51 years (4); ^H^ vocational or primary school (1), secondary school (2), higher education (university) (3); ^I^ village (1), city up to 50,000 inhabitants (2), city between 50,001 and 100,000 inhabitants (3), city between 100,001 and 500,000 inhabitants (4), city over 500,000 inhabitants (5); ^J^ very good (4), good (3), not good not bad (2), bad and very bad (1).

**Table 5 foods-11-00830-t005:** Values of dominants and medians for variables describing the reasons for not consuming wild game and for sociodemographic variables.

Behavior of Respondents/Sociodemographic Variables	Cluster 1	Cluster 2	Cluster 3	Cluster 4
Number of respondents (*n*)	197	82	162	73
Little availability of wild game ^A^	1	2	1	1
High price of wild game ^A^	1	2	1	1
Fear of disease ^A^	1	1	1	2
No family tradition of eating game ^A^	1	2	1	1
Lack of skills in preparing a tasty dish ^A^	1	1	1	1
Unacceptable taste of meat ^A^	1	1	1	1
I believe that good game dishes can only be eaten in the restaurant ^A^	1	1	1	1
I am not convinced that this meat is healthier than other meat ^A^	1	1	2	1
Ethical aspects ^A^	1	1	1	1
Other ^A^	1	1	1	1
Gender ^B^	2	2	2	2
Age ^C^	1	3	2	1
Education ^D^	2	2	2	2
Dwelling place ^E^	2	2	3	2
Financial situation ^F^	3	2	3	3

^A^ yes (2), no (1); ^B^ women (2), men (1); ^C^ 1 (18–13 years), 2 (31–40 years), 3 (41–50 years), 4 (over 51 years); ^D^ vocational or primary school (1), secondary school (2), higher education (university (3); ^E^ village (1), city up to 50,000 inhabitants (2), city between 50,001 and 100,000 inhabitants (3), city between 100,001 and 500,000 inhabitants (4), city over 500,000 inhabitants (5); ^F^ very good (4), good(3), not good not bad (2), bad and very bad (1).

## Data Availability

The data presented in this article are available on reasonable request, from the corresponding author.

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
