# Peer review of "Consumers’ Perception and Preference for the Consumption of Wild Game Meat among Adults in Poland"

_foods, 2022, doi:10.3390/foods11060830_

Round 1

Reviewer 1 Report

I would suggest-

to add more studies and surveys from 2020 and 2021,

-more sources from WoS and Scopus databases from 2020 and 2021,

-statistical methods should be described  more deeply.

-also to add the limits and barriers of the research.

Author Response

Dear Reviewer 1,

Thank you for reviewing our manuscript. Revisions were made according to the reviewers’ comments. Changes in the manuscript are indicated in Track revisions.

We look forward to your response.

I would suggest-

to add more studies and surveys from 2020 and 2021,

-more sources from WoS and Scopus databases from 2020 and 2021,

Response: According to your suggestion, we have added more new publications. Please note that some publications (e.g. 82) present data from experiments that were not repeated later, but the information remains valid.

-statistical methods should be described more deeply.

Response: Following your suggestion, we added new information as below:

The Mann-Whitney U test was used to assess the influence of eating / not eating wild game on sociodemographic factors. The null hypothesis is the lack of significant differences between the distributions of factors in both groups, the alternative hypothesis - significant differentiation of factors caused by eating wild game. The significance of the test was assumed at the level of p £ 0.05

-also to add the limits and barriers of the research.

Response: Thank you for your suggestion we describe the limitation of our study and we added barriers.

 Authors

Reviewer 2 Report

  1. There is too much references about nutritional value of game meat but this paper is oriented to consumer behaviour. There is a lack of studies related to consumers behaviour in game meat purchase and behaviour, so as studies about imortance of consumer segmentation.
  2. Why and for which variables authors used Cronbach alpha, it is not easy to undestand.
  3. Which variables (you mentioned 69 for eaters and 10 for non-eaters) are used for Cluster analysis?
  4. Authors should check if eaters and non-eaters differ according to sociodemographic characteristics?
  5. Reasons for and agains consumption should be presented in Figures.
  6. Discussion is too long and many just repetition of results. it should be more concrete and more oriented to discussion, not only results.
  7. In Table 2 you mentioned wild game consumption and meals prepared from it and there is many respondents who choose others as answer but you didn't mention any meals from those categories.
  8. 8. You have term attitudes in title, but how did you measure attitudes and where are items for attitudes maesurement in Methodology section?

Author Response

Dear Reviewer 2,

Thank you for reviewing our manuscript. Revisions were made according to the reviewers’ comments. Changes in the manuscript are indicated in Track revisions.

We look forward to your response.

  1. There is too much references about nutritional value of game meat but this paper is oriented to consumer behaviour. There is a lack of studies related to consumers behaviour in game meat purchase and behaviour, so as studies about importance of consumer segmentation.

Response: Thank you for pointing it out. We agree with the Reviewer. According to your suggestion, we added more information about consumer behaviour and consumer segmentation. The paragraph on the nutritional value of game meat has been shortened.

Why and for which variables authors used Cronbach alpha, it is not easy to understand.

Response: Thank you for your question. We added more information in section "Data Analysis" lines as below:

Before starting the analysis of the survey results, it is necessary to verify the correctness of the survey questionnaire structure. Checking its internal consistency and reliability. For this purpose, the Cronbach's alpha test was used. The value of the alpha coefficient for the entire questionnaire (all questions in the survey) is 0.79. This result indicates the satisfactory consistency of the survey questionnaire and allows the obtained results to be used in further analysis.

  1. Which variables (you mentioned 69 for eaters and 10 for non-eaters) are used for Cluster analysis?

Response: Thank you for your question. We added new information as below:

In the analysis of game eaters, 69 variables describing the attitude of game eaters to this meat and its products were used (answers to questions 3-10 in the questionnaire).

  1. Authors should check if eaters and non-eaters differ according to sociodemographic characteristics?

Response: Thank you for your question. We added new information as below:

It can be assumed that eating or not eating wild game can significantly differentiate the respondents. To verify this, the chi-square test was performed for the influence of gender of the respondents (feature on a nominal scale) on game consumption and the Mann-Whitney U test for the remaining sociodemographic factors (feature on an ordinal scale). The results of the Mann-Whitney U test and median values in both groups are shown in Table 2.

Table 2. Values of test statistics and significance p in the Mann-Whitney U test and median values in the group of people consuming and not consuming game.

Population

Features

Mann-Whitney U test

Medians

test statistic U

p

no eating wild game

eating wild game

Age (years)

164762.5

0.000

18–30

31-40

Education

137940.0

0.000

secondary school

higher education

Dwelling place

187441.5

0.609

city up to 50,000 inhabitants

city up to 50,000 inhabitants

Financial situation in own opinion

142491.0

0.000

good

good

Based on the chi-square test (c2=42.341 p=0.000), it can be concluded that gender significantly differentiates the propensity to eat wild game. Similarly, on the basis of the M-W U test, it can be stated for age, education and financial situation. People who do not eat game are mainly younger women, aged 18-30, with secondary education. The average game eater is a middle-aged man (31-40 years old) with a university degree. The financial situation in both groups is similar (the medians have the same value), but people who eat game assess their financial situation on average slightly better.

  1. Reasons for and against consumption should be presented in Figures.

Response: Thanks for the suggestion, but we have a long article with five tables and two figures. Therefore, these data are presented in the text.

  1. Discussion is too long and many just repetition of results. it should be more concrete and more oriented to discussion, not only results.

Response: The discussion has been improved and completed in accordance with the suggestions. Thank you for your valuable comments. We removed our results from the discussion section.

  1. In Table 2 you mentioned wild game consumption and meals prepared from it and there is many respondents who choose others as answer but you didn't mention any meals from those categories.
  2.  You have term attitudes in title, but how did you measure attitudes and where are items for attitudes measurement in Methodology section?

Response: Thank you for pointing this out. We did not test the attitudes.

The title was changed. It sounds like now: Consumers' Perception and Preference for the Consumption of Wild Game Meat among Adults in Poland

 Authors

Reviewer 3 Report

The study by Czarniecka-Skubina et al., was aimed at investigating preferences and opinions of Polish consumers regarding game meat, its use in their diet, and to identify consumer segments. The study is weel executed and results clearly presented.

Minor concerns are reported below:

Title: ‘Consumers' Perceptions, Attitudes and Preferences for the Consumption of Wild Game Meat among Adults in Poland’ should be ‘Consumers' Perception, Attitude and Preference for the Consumption of Wild Game Meat among Adults in Poland’

In the introduction section in not necessary to insert the paragraph’s title ‘2. Literature review’

Line 133: was the study approved by an Ethical committee? Did the participants provided a signed informed consent to participate?

Questionnaire: more information should be provided to have a clearer idea about items and scales used. Could be the questionnaire provided as supplementary material?

Author Response

Dear Reviewer 3,

Thank you for reviewing our manuscript. Revisions were made according to the reviewers’ comments. Changes in the manuscript are indicated in Track revisions.

We look forward to your response.

The study by Czarniecka-Skubina et al., was aimed at investigating preferences and opinions of Polish consumers regarding game meat, its use in their diet, and to identify consumer segments. The study is weel executed and results clearly presented.

Response: We appreciate your positive opinion and valuable comments

Minor concerns are reported below:

Title: ‘Consumers' Perceptions, Attitudes and Preferences for the Consumption of Wild Game Meat among Adults in Poland’ should be ‘Consumers' Perception, Attitude and Preference for the Consumption of Wild Game Meat among Adults in Poland’.

Response: Thank you for paying attention. According to your suggestion, the title was changed. It sounds like now: Consumers' Perception and Preference for the Consumption of Wild Game Meat among Adults in Poland.

 In the introduction section in not necessary to insert the paragraph’s title ‘2. Literature review’

Response: Thank you for pointing this out. It has been deleted.

Line 133: was the study approved by an Ethical committee? Did the participants provided a signed informed consent to participate?

Response: Thank you for your question. This information has been added in section:

  1. Materials and Methods; 2.1. Study Design and Participants.

We added new information like below:

Considering the anonymous nature of the online survey and the inability to track the sensitive personal data of respondents, the survey did not require the consent of the ethics committee (as well, written informed consent for the study was not required).

 Questionnaire: more information should be provided to have a clearer idea about items and scales used. Could be the questionnaire provided as supplementary material?

Response: Thank you for pointing this out. The questionnaire was added as supplementary material.

Authors
